# Temporal Consistent Semantic Video Color Transfer from Multiple References

## Abstract

*Transferring the color from aesthetically high-quality reference color content to captured unpleasant color content is required for the media and entertainment industry. The expert color artists manually change and edit the color tone of unpleasant color content so that it becomes aesthetically pleasing and matches with the other scenes of the main content. Inspired by the style transfer works, the photorealistic color transfer approaches aim to transfer color and brightness from the reference or style video to the main content video. However, those approaches face significant challenges due to induced color artifacts in the final output, computationally expensive, and lacking semantic correspondence. In this work, we propose a temporally consistent semantic video color transfer approach that not only overcomes existing limitations of the color transfer approaches but provides flexibility to the colorist while performing color grading in studios. The temporal inconsistency due to temporally inconsistent semantic information incorporation is handled by an online training approach to make the output temporally consistent. A quantitative comparison shows the effectiveness of our approach as compared to existing solutions. We also perform extensive subjective analysis to showcase the shortcomings of existing solutions and how our solution addresses this.*

## 1. Introduction

The color editing of video content has become an essential tool for the multimedia industry. The expert colorists in studios perform color grading for several reasons. One of the main reasons is due to aesthetically or artistically unpleasing colors in captured content. Along with this, with the wide deployment of mobile devices, it is very often that multiple users capture their videos in the same event (such as concerts and sports), or a user uses multiple cameras to shoot from multiple angles under the same scene (such as cooking shows, teaching video to repair cars). In those scenarios, the color and brightness level of the same content may change due to capturing from multiple perspectives

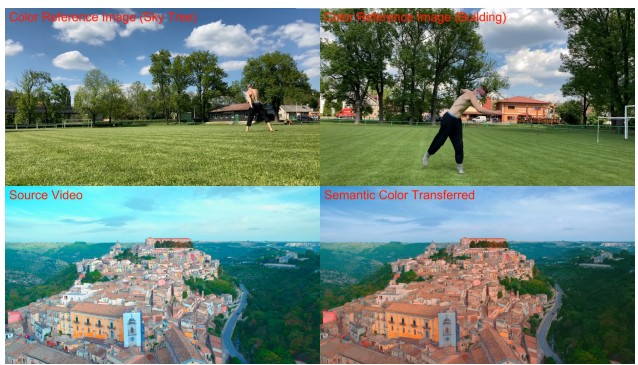

Figure 1. The main objective achieved by our approach is shown here. The source video is the input video whose color is to be transformed. The user has the references (as shown in the top row), and the user gives the information about the color from which semantic regions are to be transferred (shown in red-colored font). Our proposed approach transfers the color from that respective semantic region of references and produces the semantic color transferred output. The full video is in https://anonymous.4open.science/r/video-color-transfer-5023.

and the difference in rendering of scenes from each camera's pipeline. Another common scenario is outdoor video shooting, where the lighting conditions may change depending on the environmental conditions, which produces varying color and brightness levels in the captured footage. To have a good immersive experience of perceiving content, the color and brightness levels should be consistent across different perspectives. The switching/ transition between different perspectives should be smooth and should not feel different. There can also be many other scenarios in the multimedia entertainment industry where the color and brightness levels between multiple scenes should be consistent to give the same perceptual experience.

To address these issues, photorealistic style transfer has emerged as a new research domain. It is a subset of style transfer, where both the style image and the content image are real-world photos, and it is expected that the output style transferred image will be a photo-realistic image. Unlike style transfer approaches, which transfer textures, the pho-

torealistic style transfer methods aim to transfer color and brightness from style image to content image. The style transfer works mostly inspire the existing photorealistic style transfer and use a combination of VGG features [18] and Zero-phase Component Analysis (ZCA) [12]. Existing works mainly use the deeper VGG encoder-like style transfer works and focus on training the decoder. After that, they use ZCA to transfer color and brightness. Unlike other studies that consider even deeper VGG features, we use the first two layers of VGG architecture and the results showed that it is enough to transfer the color and brightness information. If we use the features from the deeper layer, those features will contain the texture information along with the color and brightness information. In that scenario, the WCT will try to transfer that textural information from the style image, and it is undesirable. Due to this, most of the existing ZCA-based photorealistic style transfer methods face the challenges of artifacts in the final output and out-of-memory issues due to computationally expensive VGG architecture. Our findings with shallow VGG features overcome both issues. We also observed that we do not need multiple ZCA blocks; one ZCA is enough for color and brightness.

In this document, we propose a novel algorithm for semantic-wise color transformation for videos and overcome issues like temporal inconsistency that generally arise when we use semantic mask incorporation during color transfer. We perform color (without distorting the textures) transfer from selected semantic regions of style (reference) video/image to the selected semantic regions of source videos. The semantic region is defined as the portions of an image or video that belong to the same group of semantic categories, like tree, sky, sea, etc. The semantic color/brightness transfer means changing the sky/sea/ tree regions of a video (content video) to be similar to the same semantic region in another video (style/ reference video). In our approach, the user will give input about the semantic class label of the reference image, whose color will be transferred, and the semantic class label of the content image, where the color will be replaced. The user will be given the freedom to choose the semantic region of interest in the style image and the semantic region of interest in the content image. For example, the user can select to transfer the color from the sky in the style image to the color of the river in the content image. If no user input is given, the same semantic labels will be matched between the style image and the content image, and color will be transferred semantic region-wise. It means the sky from the style image will be matched to the sky in the content image and the same for other semantic regions. If there is a semantic label in the content image that is not present in the style image, no color transfer will be performed. Figure 1 shows an output example of our color transfer algorithm, which explains what we achieved in this work.

The main key contributions to this work are as follows:

- **Semantic color transfer form multiple references:** This work proposes the first semantic-wise color transfer algorithm on videos. This algorithm will take multiple images/videos as input. Users can select which semantic region of a content video to modify by using the color from the semantic region selected in the reference style video. This algorithm will perform the transformation of that semantic region.
- **Flexible transfer via modular design:** The proposed framework allows users to group semantic classes into a super-class or allows users to divide a semantic class into sub-classes and use the new user-defined semantic class/classes to perform the transfer. The proposed framework can incorporate different segmentation methods.
- **Temporal stability via online training:** The temporal inconsistency in segmentation masks (e.g., pixels in the same classes along the time domain are classified into different classes) in the case of videos creates flickering artifacts in the output videos. We propose a novel technique to handle the temporal inconsistency in the segmentation masks through online training.
- **Lightweight:** Unlike other works in color transfer, which deploy big models, we use a shallow architecture for transferring color. We proved that a shallow architecture is more suitable for this task.

## 2. Prior Art

Photorealistic style transfer is a sub-field of style transfer that mainly aims to transfer color and brightness without distorting textures present in the content image and produce a photorealistic image. The Whitening Coloring Transform (WCT) [12] is a popular style transfer technique that mainly influences recent developments in photorealistic style transfer. In WCT, the VGG features [18] are used to extract image features, and the Zero-phase Component Analysis (ZCA) block performs the transformation in the feature domain to transform the content features so that features become like style/reference features. After that, WCT2 [22], PhotoWCT [13], PhotoWCT2 [7], and PCA [6] worked in that direction, and they mostly focus on designing different kinds of architecture and training mechanisms to train a network whose encoder is VGG pre-trained (locked) weights and whose decoder is randomly initialized network parameters which will be trained (fine-tuned) later on the target application. Their main focus was on training that deep encoder and decoder network in such a way that it does not distort the textures of the input and it can be reconstructed without any distortion. By doing so, those works achieve photorealistic style transfer using the concepts of style transfer. Those approaches use ZCA in multiple levels of the trained architecture to get a good transformation. PhotoNAS [2] developed an architecture search method to de-

velop a lightweight architecture. NLUT [3] and IPST [15] developed a test-time training approach with VGG features to perform color transfer. Bilateral [21] estimates region-wise parameters and perform local edge-aware affine transform to transfer colors. In a similar direction, the Neural-Preset [11] estimates the color transformation parameters for photorealistic color transfer. AdaCM proposed a multi-layer perceptron (MLP) based mechanism and trained an MLP to transfer the colors of the content image using the reference guidance.

## 3. Multiple Reference Driven Temporally Consistent Semantic Video Color Transfer

Our proposed reference-driven temporally consistent semantic video color transfer consists of multiple modules. We will discuss the overall pipeline before diving into detailed descriptions of modules.

### 3.1. The Whole Color Grading Pipeline

Figure 2 shows the whole color transfer or color grading pipeline. The input content video is a captured scene with bad colors (not aesthetically pleasing) or the colors that the user wants to change. Multiple style references contain the scene whose color is aesthetically pleasing and the objective is to transfer the color semantic-wise from multiple style references to content video. The multiple style references can be videos or images. Now, the user needs to provide the requirement about the color from which the semantic region of a style reference will be transferred to the content video. If no user input is given, all the different semantic regions in the content video will be considered for color transfer from the same semantic region of references.

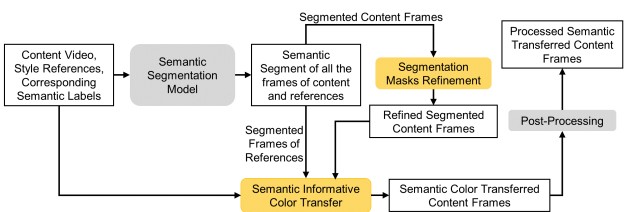

Figure 2. Reference driven semantic color transfer pipeline

At first, all the frames of the content video and the style references are segmented. After that, the segmented content frames are passed to the segmentation mask refinement module to make the final output temporally consistent. The temporal inconsistency arises due to the semantic segmentation model. Now, the refined segmented content frames and the segmented reference frames are passed to the semantic informative color transfer module for semantic-wise color transfer. In the end, the guided filtering stage is used as a post-processing stage to improve the semantic boundary regions of semantic color-transferred content frames and pro-

duce the final output.

Algorithm 1 shows the algorithm description of the pipeline. $\mathbb{V}_C$ is the input content video and $\mathbb{V}_{CS}$ is the final color transferred video. The style references $I_{S_j}$ can be both images and video. In the case of video references, we only consider frames with a predefined skip to extract the semantic-wise color features. We do not consider all the frames as consecutive frames remain almost similar content and semantic regions. Therefore, it will create redundancy. $L_{S_j}$ contains semantic labels of style reference $j$, and the color from that semantic region will be transferred to the semantic region $L_{C_j}$ of the content video.

---

**Algorithm 1:** Video color transfer algorithm

---

**Result:** Semantic color transferred video $\mathbb{V}_{CS}$
**Data:** Style references $I_{S_j} \in \mathbb{R}^{H_S \times W_S \times 3 \times P_S}$,
where $j = 1, 2, 3, ...T$, $P_S = 1$ for image;
Content video $\mathbb{V}_C \in \mathbb{R}^{H_C \times W_C \times 3 \times P_C}$;
$L_{S_j}$: semantic labels to be selected from style references, $L_S = \{L_{S_1}, L_{S_2}, L_{S_3}, ...L_{S_T}\}$;
$L_{C_j}$: corresponding semantic labels of each style label $L_{S_j}$ to the semantic label in content video $\mathbb{V}_C$;
$T$: number of style references;
**Semantic Segmentation:**
$B_{S_j} = F_{seg}(I_{S_j}; \theta_{F_{seg}}), B_{S_j} \in \mathbb{R}^{H_S \times W_S \times P_S}$
$B_C = F_{seg}(\mathbb{V}_C; \theta_{F_{seg}}), B_C \in \mathbb{R}^{H_C \times W_C \times P_C}$
**Segmentation Mask Refinement:**
$M_C = F_r(\mathbb{V}_C, B_C; \theta_{F_r})), M_C \in \mathbb{R}^{H_C \times W_C \times N \times P_C}$
**Semantic Color Transfer:**
$f_{S_j} = F_{enc_1}(I_{S_j}), \forall j$
**for** $p$ *in all* $P_C$ *frames* **do**
   $f_C{}^p = F_{enc_1}(\mathbb{V}_C{}^p)$;
   $M_C{}^p = M_C[p]$;
   $\hat{f_{CS}}{}^p = 0$;
   **for** $j = 1, 2, ..., T$ **do**
      **for** $k^{th}$ *semantic label in* $L_{S_j}$ **do**
         $f_{S_{k_j}} = f_{S_j}[B_{S_j} == L_{S_j}[k]]$;
         $f_{CS_k}{}^p = ZCA(f_C{}^p, f_{S_{k_j}})$
         $\hat{f_{CS}}{}^p += f_{CS_k}{}^p \times M_C{}^p[L_{C_j}[k]]$
      **end**
   **end**
   **for** $k^{th}$ *semantic label not in* $L_S$ **do**
      $f_{CS_k}{}^p = f_C{}^p$;
      $\hat{f_{CS}}{}^p += f_{CS_k}{}^p \times M_C{}^p[k]$;
   **end**
   $\hat{\mathbb{V}_{CS}}{}^p = F_{dec}(F_{enc_2}(\hat{f_{CS}}{}^p))$;
   $\mathbb{V}_{CS}{}^p = \mathcal{GF}(\hat{\mathbb{V}_{CS}}{}^p, \mathbb{V}_C{}^p)$
**end**
Combine all frames $\mathbb{V}_{CS}{}^p \rightarrow \mathbb{V}_{CS}$

---

$F_{seg}$ is a pre-trained semantic segmentation model that will divide each content frame and style reference frame

into multiple semantic regions. $B_C$ is the semantic segmentation map of content all the content frames, where each pixel location stores the semantic class label where that pixel belongs. The same is true for style references, where $B_{S_j}$ carries the pixel-wise semantic labels of style reference $j$. The content segmentation $B_C$ is not temporally consistent and has a lot of flickering in the semantic boundary region. We observed flickering in the final output when we used the $B_C$ without any further processing. Therefore, we develop a segmentation mask refinement approach where we retrain another model $F_r$, which is trained to map the input $\mathbb{V}_C$ into the output $B_C$. After training, we perform inference and get the pixel-wise semantic class-wise probability map and it is used for semantic color transfer without any flickering. In the semantic color transfer module, we perform the color transfer using all the inputs and the processed segmentation maps. Even though we get rid of the flickering, we observe mild ghosting artifacts in the semantic boundary region of the final output. We use the guided filter [10] $\mathcal{GF}$ to overcome this issue and produce the final output without any artifacts.

### 3.2. Segmentation Mask Refinement

This module is required when we want to transfer the color from the content video instead of the content image. We perform the semantic segmentation on each video frame independently. Therefore, the segmentation masks are inconsistent in the temporal domain, and it eventually creates flickering artifacts in the output video. We develop a mechanism to finetune the segmentation masks to improve temporal consistency and as a result, it removes the flickering artifacts from the output video. The finetuning of the segmentation mask is performed via online training. As this training will be performed during the color transfer, it is termed as an online training. It is mathematically expressed as $M_C = F_r(\mathbb{V}_C, B_C; \theta_{F_r})$, as shown in Algorithm 1.

During online training, we use $\mathbb{V}_C$, $B_C$ to train the parameters $\theta_{F_r}$ of $F_r$ to improve temporal consistency and remove the flickering from the final output. $B_C$ contains the semantic segmentation class labels that are generated using pre-trained model $F_{seg}$. We broadcast the $B_C$ so that each pixel has one hot encoding and $B_C$ becomes $B_C' \in \mathbb{R}^{H_C \times W_C \times N \times P_C}$, where $N$ is the number of semantic classes. We design another training and inferencing framework, which will finetune $B_C'$ during the color transfer on a single video. The idea is to train another neural network $F_r$ with learnable parameter $\theta_{F_r}$, to learn the mapping between $\mathbb{V}_C$ and $B_C'$. The forward pass through $F_r$ is defined as, $M_C' = F_r(\mathbb{V}_C; \theta_{F_r})$. The objective is to minimiz the distance between $M_C'$ and $B_C'$ to optimize the parameters of $F_r$.

Now, the core idea is that the network $F_r$ should not be trained for longer epochs and do not overfit the model on $\mathbb{V}_C$, $B_C'$ pair. As we train for longer epochs, the network will exactly learn $B_C'$. The estimated segmentation mask $B_C'$ of whole video data $\mathbb{V}_C$ contains flickering, and it is a high-frequency component. When we try to fit the data into the network $F_r$, the network will learn the mapping between $\mathbb{V}_C$ and $B_C'$ using the low-frequency and mid-frequency details during the initial epochs of learning. Now, if we train longer, the high-frequency flickering will be captured. Therefore, training for a few numbers of epochs will help to achieve $M_C'$ without flickering. After training, the trained model $F_r$ is used to generate the segmentation masks without temporal inconsistency. It is mathematically expressed as, $M_C = F_r(\mathbb{V}_C; \theta_{F_r})$.

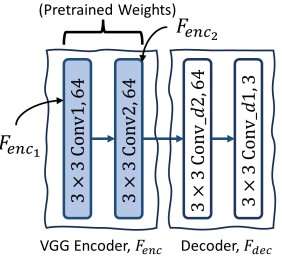

Figure 3. Our proposed lightweight model for color transfer. In $F_{enc_1}$, $[3 \times 3 \ Conv1, 64]$ means $Conv1$ block has $3 \times 3$ convolutional kernels and $64$ output features.

### 3.3. Lightweight Semantic Color Transfer

The semantic color transfer has two main components. The first one is the main architecture and the second one is the color transfer algorithm using semantic maps with labels.

**The lightweight color transfer network:** The color transfer network has only four consecutive convolution layers with ReLU non-linearity after each convolution layer, as shown in Figure 3. We term the first two layers as encoder and second two layers as decoder. The encoder is initialized with the pre-trained weights of the first two layers of VGG architecture. The semantic informative ZCA module works on the extracted features of the first convolution layer of the encoder $F_{enc_1}$ and transfers the content features based on reference/style features. After that, the second layer of encoder $F_{enc_2}$ processes the transferred content features. The decoder $F_{dec}$ reconstructs the final color transferred output image. As the encoder layers are initialized with pre-trained weights, we only train the decoder $F_{dec}$.

**The color transfer algorithm:** The color transfer process is performed in each frame independently, as shown in Algorithm 1. The features are extracted from the first layer of encoder $F_{enc_1}$ for both input and references. $f_{S_j}$ is the extracted features of $j^{th}$ reference. If the reference is video, it will be the features extracted from different frames. Now, for $p^{th}$ frame, $f_C^p$ is the extracted image features from $p^{th}$ frame of $\mathbb{V}_C^p$ and $M_C^p$ is the corresponding refined segmentation mask probability maps. The provided user input

is $L_{S_j}$ and $L_{C_j}$. The color from semantic region $L_{S_j}$ from $j^{th}$ reference image will be transferred to the semantic region $L_{C_j}$ of the content video. Now, for the $k^{th}$ semantic label present in $L_{S_j}$, the features from $f_{S_j}$ will be picked from those pixel locations where the semantic label is $L_{S_j}[k]$ and the picked feature is $f_{S_{k_j}}$. ZCA will transform the content features $f_C{}^p$ using $f_{S_{k_j}}$ and the transformed feature is $f_{CS_k}{}^p$. Now the pixel-wise probability map of $p^{th}$ content frame for the semantic class $L_{C_j}[k]$ is $M_C{}^p[L_{C_j}[k]]$ and it is used to get semantic feature filtering of transformed feature is $f_{CS_k}{}^p$. Like this, we perform the operation for all the semantic labels and accumulate the semantic-driven transformed features $\hat{f_{CS}}{}^p$. If the user has not mentioned some semantic labels, we do not perform any ZCA transformation. Finally, the second layer of encoder $F_{enc_2}$ and decoder $F_{dec}$ are used to reconstruct the tarnsformed frame $\hat{\mathbb{V}_{CS}}{}^p$. After that, guided filtering $\mathcal{GF}$ is used to improve the semantic boundary regions.

### 3.4. Speeding Up Tricks

The computational burden whole inferencing process can be improved by following a few things without losing the perceptual visual quality. The tricks for faster inference that can be adopted in different scenarios are as follows:

- **Segmentation mask estimation model via spatial resolution resampling:** To process the high-resolution image, the segmentation mask will take a lot of inference time. We can downscale the spatial resolution of reference and content frames to perform the semantic segmentation and, after that, upscale the semantic mask using the conventional interpolation technique. It will reduce the inference speed and will also increase the inaccuracy in boundaries. Those inaccuracies in boundary regions can be handled by the guided filter itself.
- **Transferring color from reference videos via temporal resampling:** In this case, we can skip a few intermediate frames to reduce the computational overhead. The consecutive frames contain almost the same information, therefore, there is no advantage to consider all the frames of the reference videos.
- **Color transfer network via lower spatial resolution of style references:** In the Color and Brightness Transfer network, we can use downscaled reference video frames without any perceptual quality change. In this network, the content video frames will be processed in the actual resolution of the frame, as we will lose textural information if we downscale it. On the other hand, we need reference video frames for color and brightness information only, and that information remains intact even with downscaling since ZCA works in the VGG feature domain.
- **Online training of segmentation mask refinement model via lower spatial resolution:** The segmentation mask refinement model can be trained and tested on both

downscaled images and segmentation masks without any perceptual change in the final output video.
- **Guided filtering:** To speed up the guided filtering process, we adopt the fast guided filtering [9] where the image is subsampled $N$ times to calculate the guided filter parameters. In our experiment, we use $N = 4$ to calculate guided filter parameters.

## 4. Results and Discussions

### 4.1. Experimental Setup

#### 4.1.1. Datasets

We use randomly selected 5000 images from the MS-COCO dataset [14] to train the color transfer network. However, any natural images can be used to train this network. We create a test dataset, Landscape100, for quantitative testing by collecting 100 random pairs of images from landscape dataset [1], wherein each pair, one is the content image, and another one is a reference image. Images for subjective analysis are collected from multiple sources [6, 8, 15, 16]. We also use the Inter4K dataset [19] as a test dataset, which consists of 1000 high-quality videos.

#### 4.1.2. Training and Inference Details

**Color Transfer Network:** The decoder of color-transferred network $F_{dec}$ is trained to reconstruct the input image $x$ to the network. The decoder gets the extracted image features from the encoder $\{F_{enc_1}, F_{enc_2}\}$ and reconstructs the input image $x$. We train the model using randomly cropped $256 \times 256$ image patches for 200 epochs; each epoch consists of 1000 batch update, and batch size is 16. Adam optimizer with learning rate $10^{-4}$ is used to update the parameters of decoder $F_{dec}$. The mean-squared error loss function is used to calculate the errors between input $x$ and reconstructed output.

**Semantic Segmentation Model:** We use the state-of-the-art semantic segmentation model, Mask2Former [4, 5] to perform semantic segmentation of both content video frames and references. The Mask2Former model is trained on ADE20K dataset [23, 24] that contain around 250 classes. We merge the different sub-classes into multiple super-classes categories for simplicity. We merge all the smaller sub-classes into 7 different super-classes. Those super-classes are as follows: Stationary man-made outdoor objects, Non-stationary man-made outdoor objects, Indoor objects, Sky, Trees, Natural stationary objects (Earth, Mountain, Field, and Ground), and water bodies.

**Segmentation Mask Refinement Model:** We use the U-Net architecture [17] as semantic mask refinement model $F_r$. However, unlike the official U-Net architecture, we do not use the batch normalization layer as batch normalization helps to converge very fast and overfits the test datasets. Our objective is not to overfit the model on the test content video

frames and semantic masks pair completely as overfitting captures the high frequency flickering. The batch normalization helps in faster overfitting and therefore, getting the stopping point becomes a tedious task. Our requirement is to get a trained model that will remove flickering and keep the segmentation loss minimum. In our experiment, for a $4K$ video with 300 frames, we experimentally found that training 30 epochs gives us the desired output for a wide range of test videos. We also observed that the performance does not change with the number of epochs change $\pm 5$. Therefore, there is a saddle region where we can stop training the model To train the semantic mask refinement model $F_r$, we use the mean-squared error as a loss function, and the Adam optimizer with learning rate $10^{-4}$ is used as a weight update rule.

| | WCT2 | Bilateral | NLUT | IPST | PhotoWCT2 | PCA | Ours |
|---|---|---|---|---|---|---|---|
| niqe | 2.901 | 2.583 | *2.645* | 3.008 | **2.550** | 2.65 | **2.550** |
| piqe | **30.73** | 35.85 | 36.56 | 34.59 | 34.63 | 35.26 | *33.84* |
| ssim | 0.814 | 0.872 | 0.782 | 0.883 | 0.797 | 0.820 | 0.857 |

Table 1. Quantitative analysis of our proposed approach as compared to existing approaches. The best and second best results of niqe and piqe are shown in **bold** and *italic* respectively.

| | Consistency | city | girl | kelly | monkey | night | pedestrian | stream2 | sunset |
|---|---|---|---|---|---|---|---|---|---|
| WCT2 | Short | **70** | *20* | 140 | 80 | **80** | **160** | **90** | *20* |
| | Long | **170** | 90 | 490 | 300 | **70** | **320** | **130** | *20* |
| Bilateral | Short | 220 | 30 | *70* | 60 | 160 | 290 | 150 | 40 |
| | Long | 440 | 190 | *200* | 290 | 70 | 510 | 200 | 60 |
| NLUT | Short | 160 | 20 | 140 | 100 | 270 | 270 | 160 | 110 |
| | Long | 320 | 110 | 320 | 390 | 190 | 540 | 210 | 190 |
| IPST | Short | 180 | 40 | 110 | **40** | 260 | 260 | 190 | 30 |
| | Long | 340 | 200 | 290 | **190** | 120 | 440 | 250 | 30 |
| PhotoWCT2 | Short | 120 | 20 | 80 | 100 | 160 | *190* | 130 | 30 |
| | Long | 270 | 170 | 320 | 440 | 70 | *360* | 160 | 30 |
| PCA | Short | 140 | 20 | 100 | 100 | 340 | 250 | 180 | *20* |
| | Long | 300 | 120 | 350 | 490 | 90 | 470 | 240 | *20* |
| Ours | Short | *120* | **10** | **60** | **40** | 250 | 260 | *110* | **8** |
| | Long | *260* | 110 | **220** | *210* | *80* | 490 | *140* | **6** |

Table 2. Quantitative comparison based on temporal consistency of our approach as compared to existing approaches. We use optical flow-based wrapping error to measure temporal consistency. All the values are an order of magnitude of $10^{-5}$. The best and second best results are shown in **bold** and *italic*, respectively.

## 4.2. Quantitative Analysis

We perform a quantitative analysis of our approach to compare the reconstruction performance with the existing approaches. Table 1 shows the performance comparison using metrics like niqe, piqe, and ssim on the Landscape100 test dataset. niqe and piqe are the no-reference perceptual quality assessment metrics. The lower value of those metrics signifies better perceptual quality. Our method performs better than others in niqe metric and produces comparable performance while using piqe metric. We also use the ssim as a metric to measure the structural similarity between the input image and the color-transferred image. We calculate the ssim on L channel of Lab color space. ZCA-based feature transforms approaches like WCT2, PhotoWCT2, and PCA use deep features for color transfer, which eventually distorts the structures in the input image and leads to

lower ssim. NLUT also faces similar challenges. In spite of being a feature transform-based algorithm, we produce higher structural similarity as compared to others. This happened as we used the shallow features from VGG. Those features are mainly influenced by low-frequency information like colors, and it does not contain high-level textural information. Therefore, we observe less structural distortion as compared to similar approaches.

We also comapred temporal consistency of our proposed approach as compared to existing approaches. Table 2 shows the performance of temporal consistency. We use Consistency Error ($CE$) as an evaluation metric, and it is defined as

$$CE(I_i, I_j) = MSE(I_i, \mathcal{M}_{i,j}, \mathcal{W}_{i,j}(I_j)). \quad (1)$$

$I_i$ and $I_j$ are the $i^{th}$ and $j^{th}$ frame respectivly. Based on estimated flow using raft [20], we wrap the $I_j$ and project it into $I_i$. The $\mathcal{M}_{i,j}$ is the occultation mask, and it is used to filter out the occulted regions from error calculation. The final mean-squred error ($MSE$) between $I_i$ and wrapped image $\mathcal{W}_{i,j}(I_j)$, excluding the occulated regions $\mathcal{M}_{i,j}$ are used to measure the temporal consistency. We use the 5 frames skip between $i$ and $j$ to measure short temporal consistency and 35 frames skip for long temporal consistency. The performance is measured on 8 different videos, as provided by [3]. We can observe from the table that our proposed approach is much more temporal consistent as compared to other approaches expect WCT2. Our results are comparable as compared to WCT2. We achieved this performance in spite of incorporating the temporal inconsistent semantic segmentation mask, and it became possible due to our proposed semantic mask refinement algorithm.

## 4.3. Subjective Analysis

Figure 4 shows the subjective comparison of our proposed semantic color transfer approach as compared to existing approaches. As there is no definite quantitative metric to analyze the superiority of the approach, we perform extensive analysis on color transfer outputs to explain the superiority and exclusive features that our approach offers. There are comparative comparisons on 12 image sets in Figure 4.

In $1^{st}$ row image, most of the methods fail to adapt the colors from the reference image and produce different other colors, whereas we produce a similar perceptual scene like the reference. In $2^{nd}$ row image, the less textural information in reference influences a lot in most of the methods, where they distort texture; however, we preserve the content textures and adapt color from reference. In $3^{rd}$ row image, most of the methods can not produce a similar perceptual image like the reference and distort the image by boosting yellow color tints.

In $4^{th}$ row image, the dominant colors present in a small region of the reference image (e.g., red-yellow color um-

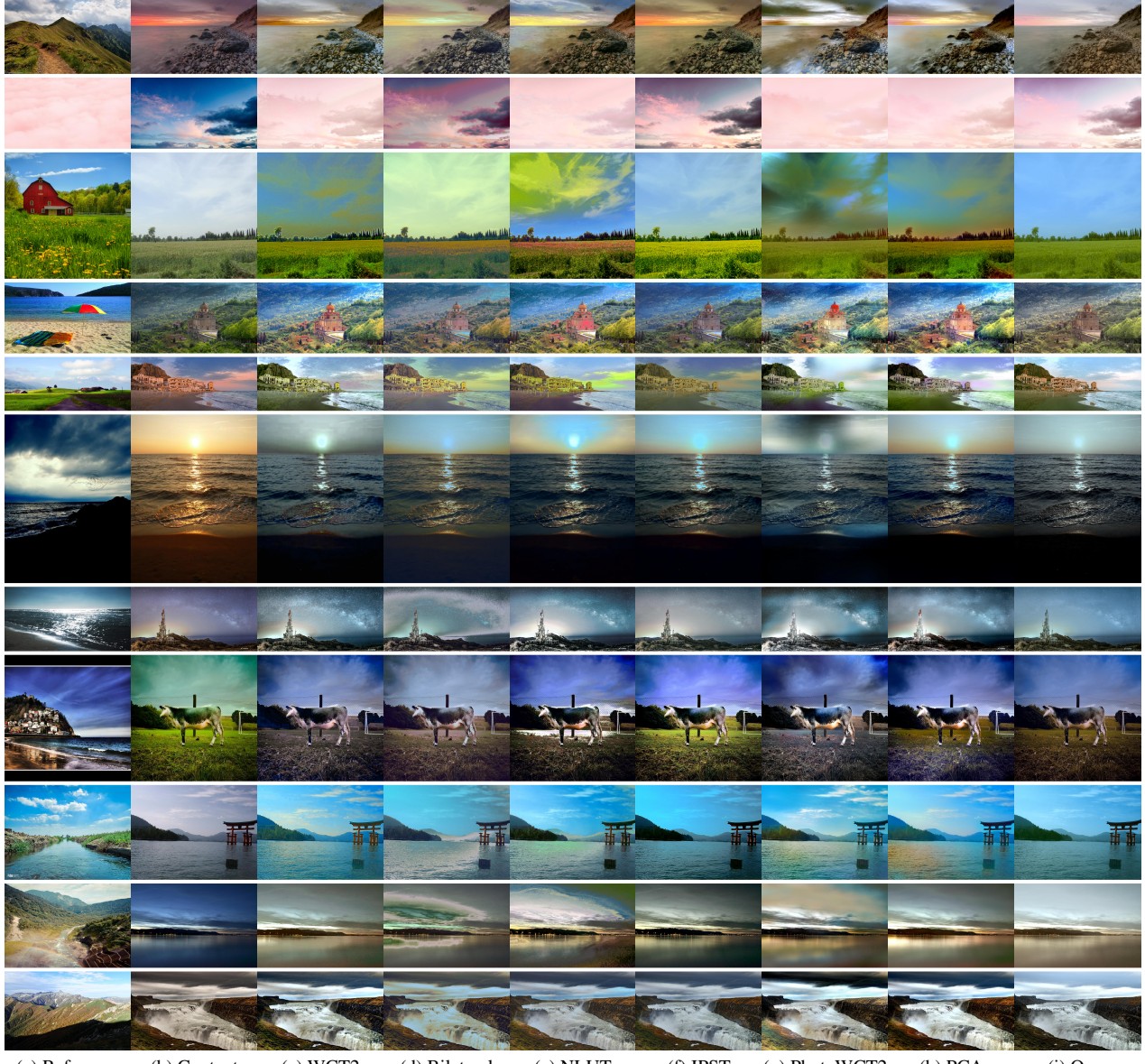

(a) Reference     (b) Content     (c) WCT2     (d) Bilateral     (e) NLUT     (f) IPST     (g) PhotoWCT2     (h) PCA     (i) Ours

Figure 4. Subjective comparison of our proposed semantic color transfer approach as compared to existing solutions.

brella) spreads and distorts the whole image for most of the methods. The results may look perceptually similar to some extent, but they are not realistic color transfer. That kind of transfer limits the use cases for professionally generated content. On the other sides, our approach produces reference like aesthetically similar content without color distortion. In $5^{th}$ row image, unlike ours, most of the methods introduce color distortions in the sky and building region.

In $6^{th}$ row image, all the methods create distortion in the sky region, and the colors do not match perceptually with reference, whereas we produce a similar perceptual region. The same is true for $7^{th}$ row image, where most of the methods increase the contrast and create halo-like distor-

tion; however, our approach produces similar contrast images like the content image and a similar color image like the reference image. The same contrast-based distortion is true for $8^{th}$ row image.

In $9^{th}$ row image, our method only captures the color of the water without any distortion. In both $10^{th}$ and $11^{th}$ row images, unlike others, we brought the similar looking sky and similar scene image without any distortions.

Overall, We produce outputs without any dominant color distortion, new color introduction-based distortion, and constant enhance based distortion. Sometimes, those distortions may give similar perceptual experiences, but those images do not look real, and they can not be used for pro-

fessional content creation. We witness those kinds of distortion in existing approaches as they mostly use deeper VGG features for color and brightness transfer. However, as we use shallow low-frequency features, our model is able to transfer low-frequency components like color and brightness, and our model is not influenced by the contrast and textures present in the reference images.

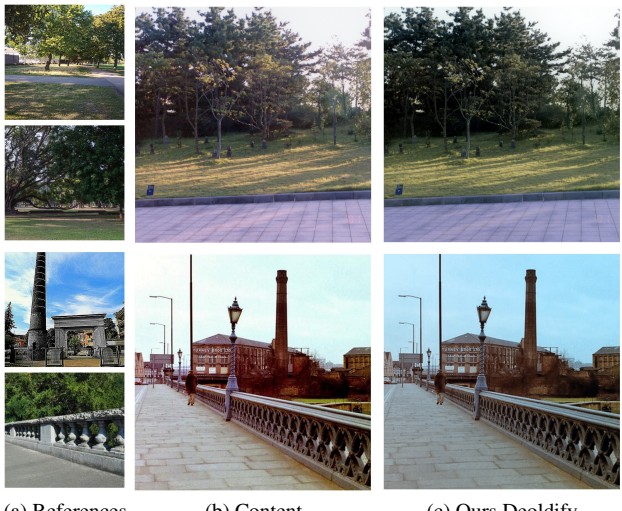

(a) References      (b) Content      (c) Ours Deoldify

Figure 5. De-oldification achieved by our proposed algorithm.

## 4.4. Additional Studies and Discussions

### 4.4.1. De-Oldification

De-oldification is a task to recover old and bad-colored content and give it a good look. Figure 5 shows the performance of our proposed semantic color transfer approach in the de-oldification scenerio. Our experimental findings show that our approach can perform de-oldification by taking semantic color input from multiple good color references.

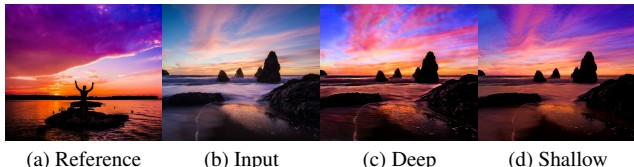

(a) Reference   (b) Input   (c) Deep   (d) Shallow

Figure 6. Comarison between Deeper vs. Shallow VGG features.

### 4.4.2. Depth of VGG

We perform experiments to showcase that shallow VGG features, as proposed in this paper, are actually useful for color transfer compared to deep VGG features. Figure 6 shows the experimental results. Unlike shallow VGG features, deep VGG features produce unwanted artifacts.

### 4.4.3. Effect of Semantic-aware Color Transfer

The semantic-aware color transfer helps to give more control to the user and perform fine-level colorization. The semantic information guides the color transfer to produce a photo-realistic output. The existing approaches mostly pick dominant colors and apply them over content. It makes the content visually attractive, but it is not photorealistic or real-looking content. With semantic-aware color transfer, we can preserve the realism of the transformed content.

### 4.4.4. Effect of Semantic Mask Refinement

As we use a semantic mask for color transfer, it produces flickering in the final output. This flickering is due to the inaccurate mask prediction and semantic label prediction in consecutive frames. In some challenging senerio, e.g., model classify the cloud from the top of the hill as a sky and sometime a ground. This misidentification leads to flickering. Our test-time refinement of semantic masks solves this issue and helps to produce a temporal consistent output. This also helps to improve semantic boundary regions where semantic masks flicker a lot. A comparative subjective study on the effect of semantic mask refinement is presented in `https://anonymous.4open.science/r/video-color-transfer-5023`.

### 4.4.5. Limitations

There are a few limitations to our proposed approach. This approach works well for color images but fails to perform the colorization of grayscale images. This is due to a fundamental building block, ZCA, which cannot map the feature distribution of grayscale images to color images. Instead of ZCA, an MLP with learnable parameters solves this problem up to a certain level. We consider it as future research, where feature transform-based semantic color transfer approaches can be used as a tool for colorization. Another challenge is the multiple colors in the same semantic regions. If there are multiple colors in the same semantic region, our approach performs the average of colors. As we use shallow features only, we observe this averaging. The color dependent sub-class division of a semantic region will help to mitigate this. We consider it as a future scope.

## 5. Conclusion

In this work, we propose a novel technique for semantic-wise transfer of the color from multiple references. We also propose an approach for making the color transfer outputs temporally consistent. Our proposed method gives flexibility to the user to choose the color from which the semantic region of a reference will be transferred to the input test video. We handled issues like textural distortion, boundary artifacts due to inaccurate segmentation masks, and temporal inconsistency. We also discussed the limitations of the existing approaches and our performance. In summary, we have achieved temporally consistent semantic-wise video color transfer from multiple reference images or videos to an input content video. This algorithm can be used as a useful tool for color-grading artists in studios.

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
