# OpenReview forum: "Temporal Consistent Semantic Video Color Transfer from Multiple References"
_thecvf.com/CVPR/2025/Workshop/CVEU — CVPR 2025_

### Official Review · Reviewer_ih9C · 2025-03-15
**the result is promising**

**Rating:** 4
**Confidence:** 4

**Review:**

This paper proposes a temporally consistent semantic video color transfer approach to improve color grading in the media and entertainment industry.

It addresses limitations of existing methods, such as color artifacts, and lack of semantic correspondence, by introducing an online training approach for temporal consistency. The method performs well, outperforming prior solutions in quantitative and subjective evaluations, and has strong industrial applications by reducing manual effort for colorists.

However, it relies heavily on input format, which may limit generalizability, and the online training process could introduce computational overhead, impacting real-time usability.

---

### Official Review · Reviewer_nBWt · 2025-03-19
**Review Comment for Temporal Consistent Semantic Video Color Transfer from Multiple References**

**Rating:** 3
**Confidence:** 4

**Review:**

**Paper Summary**

In this study, the authors proposed a temporally consistent semantic video color transfer method that not only overcomes the existing limitations of color transfer methods but also provides flexibility for colorists when performing color grading in the studio.

**Paper Strengths**

1. The proposed method is highly flexible, capable of selecting diverse styles from multiple images or videos and transferring them to the generated content.
2. The Mask Refiner strategy introduced by the authors enhances inter-frame coherence, while the Color Transfer strategy is lightweight yet effective.
3. The authors proposed numerous techniques to reduce training overhead without compromising performance.

**Weaknesses**

1. The paper is overly verbose, with simple concepts described using lengthy symbolic notations, which hinders readability. Additionally, the table formatting does not adhere to the official CVPR template.
2. Although the authors claim to surpass existing strategies, the quantitative and qualitative results show that the proposed method only achieves state-of-the-art performance in certain metrics, raising doubts about its overall effectiveness.
3. While the Whole Color Grading Pipeline is detailed and comprehensive, it appears to primarily describe existing methods, with the authors merely making minor structural modifications or deletions. (PS: I an unclear whether Workshop CVEU emphasizes novelty.)
4. The ablation study lacks validation of strategies such as Speeding Up Tricks, and the completeness needs improvement.

---

### Decision · Program_Chairs · 2025-03-25

**Decision:**

Accept

**Comment:**

Strengths: The proposed approach provides significant flexibility for semantic color transfer across videos, supported by practical techniques for maintaining temporal coherence. Additionally, the lightweight strategies introduced effectively reduce training overhead without performance loss.

Weaknesses: The manuscript suffers from verbosity and unclear notation, reducing readability, and quantitative evaluations do not convincingly demonstrate superiority across all metrics. Furthermore, the paper lacks comprehensive validation of proposed optimizations and offers limited novelty regarding existing methods.

Decision: Given the overall positive contribution and practical applicability, the paper is accepted. Authors are encouraged to address the identified weaknesses thoroughly in their camera-ready submission.